# Sex-Specific Predictors of Long-Term Mortality in Elderly Patients with Ischemic Cardiomyopathy

**DOI:** 10.3390/jcm12052012

**Published:** 2023-03-03

**Authors:** Hyun Ju Yoon, Kye Hun Kim, Nuri Lee, Hyukjin Park, Hyung Yoon Kim, Jae Yeong Cho, Youngkeun Ahn, Myung Ho Jeong

**Affiliations:** Department of Cardiology, Chonnam National University Medical School/Hospital, Gwangju 61469, Republic of Korea

**Keywords:** mortality, gender, elderly, ischemic, cardiomyopathy

## Abstract

Ischemic heart failure (HF) is one of the most common causes of morbidity and mortality in the world-wide, but sex-specific predictors of mortality in elderly patients with ischemic cardiomyopathy (ICMP) have been poorly studied. A total of 536 patients with ICMP over 65 years-old (77.8 ± 7.1 years, 283 males) were followed for a mean of 5.4 years. The development of death during clinical follow up was evaluated, and predictors of mortality were compared. Death was developed in 137 patients (25.6%); 64 females (25.3%) vs. 73 males (25.8%). Low-ejection fraction was only an independent predictor of mortality in ICMP, regardless of sex (HR 3.070 CI = 1.708–5.520 in female, HR 2.011, CI = 1.146–3.527 in male). Diabetes (HR 1.811, CI = 1.016–3.229), elevated e/e’ (HR 2.479, CI = 1.201–5.117), elevated pulmonary artery systolic pressure (HR 2.833, CI = 1.197–6.704), anemia (HR 1.860, CI = 1.025–3.373), beta blocker non-use (HR2.148, CI = 1.010–4.568), and angiotensin receptor blocker non-use (HR 2.100, CI = 1.137–3.881) were bad prognostic factors of long term mortality in female, whereas hypertension (HR 1.770, CI = 1.024–3.058), elevated Creatinine (HR 2.188, CI = 1.225–3.908), and statin non-use (HR 3.475, CI = 1.989–6.071) were predictors of mortality in males with ICMP independently. Systolic dysfunction in both sexes, diastolic dysfunction, beta blocker and angiotensin receptor blockers in female, and statins in males have important roles for long-term mortality in elderly patients with ICMP. For improving long-term survival in elderly patients with ICMP, it may be necessary to approach sex specifically.

## 1. Introduction

The prevalence of heart failure showed an increase due to the increase in the aging population. Ischemic cardiomyopathy (ICMP) is the most common cardiomyopathy. It is a condition when the heart muscle is weakened as a result of acute ischemic syndrome [1,2]. ICMP is most often presented with dilated morphology with wall motion defects and a history of previous myocardial infarction or confirmed coronary artery disease. Uncontrolled ischemia is a frequent cause of heart failure (HF) exacerbation after myocardial infarction in the case of progressive remodeled heart. Ischemic heart disease (IHD) has been the most significant cause of death in developed countries for several decades [3]. Women with IHD experience relatively worse outcomes compared to men. IHD and heart failure represent the leading causes of death in women, especially elderly people. IHD accounts for a third of all female deaths globally and affects nearly 48 million women in the United States [4,5]. A previous report noted that once women develop IHD, the risk of HF is high [6,7]. An expanded view of the multifactorial epidemiology of IHD and/or HF in women has identified important risk factors, including age, race, culture, ethnicity, socioeconomic status, lifestyle, and educational level influences that adversely impact cardiovascular outcomes. These disparities reflect our limited understanding of the sex differences in physiology, which are substantially related to lack of elderly female-specific data. Until now, there were few sex-specific predictors of ICMP. Therefore, we investigated sex-specific predictors on long-term mortality in elderly patients with ischemic cardiomyopathy (ICMP).

## 2. Experimental Section

### 2.1. Study Design and Population

The present study was a single-center retrospective observational study, and the study protocol was approved by our institutional review board (No. 2015-05-092).

From January 2007 to December 2017, a total of 1200 ischemic HF patients with echocardiographic findings with both left ventricular (LV) ejection fraction < 45% and LV end diastolic dimension > 55 mm were identified. Among them, 536 patients with ICMP more than 65 years old (77.8 ± 7.1 years, 283 males) were enrolled. In the present study, ICMP was defined as LV ejection fraction < 40% and LV end diastolic dimension > 55 mm with one or more of the following findings: (1) a history of prior myocardial infarction or revascularization, (2) more than 75% stenosis of left main or proximal left anterior descending coronary artery, or (3) more than 75% stenosis of two or more epicardial coronary arteries [8]. In this study, for the purpose of detecting predictor of mortality, we included patients with ejection fraction ranging from 40–45% (n = 53). The reasons of exclusion were as follows: (1) age < 65 years (n = 552), (2) no imaging studies for coronary artery (n = 60), (3) acute myocardial infarction (n = 14), or (4) severe aortic or mitral valve disease (n = 11), (5) others (n = 27) (Figure 1).

The development of death during clinical follow up was evaluated, and predictors of mortality according to sex were evaluated.

### 2.2. Echocardiographic Examination

Comprehensive two-dimensional and Doppler echocardiographic examinations were performed in accordance with the recommendation of the current guideline [9]. Echocardiographic images from various echocardiographic windows were obtained by using a digital ultrasonographic equipment system (Vivid 7, GE Vingmed Ultrasound, Horten, Norway). Digital cine loops were obtained for subsequent offline analysis. All of the data were analyzed by using the computerized offline software package (EchoPAC PC 6.0.0, GE Vingmed Ultrasound, Horten, Norway). Chamber quantification was performed according to the current recommendations and included the measurement of LV end-systolic and end-diastolic dimensions or volumes, interventricular septal and posterior wall thicknesses, LV mass, left atrial diameter or volume, and LVEF. Early (E) and late (A) diastolic velocities of the mitral inflow (E wave) were measured by pulsed-wave Doppler from the apical four-chamber view, with the sample volume located at the tip of the mitral leaflets. Deceleration time (DT) of the E wave was measured as the time between the peak early diastolic velocity and the point at which the steepest deceleration slope was extrapolated to the zero line. Early diastolic (e’), late diastolic (a’), and systolic (s’) velocities of the septal mitral annulus were obtained by tissue Doppler imaging in the apical four-chamber view. Right ventricular systolic pressure (RVSP) was measured by the maximal velocity of the tricuspid regurgitation jet using a modified Bernoulli’s equation.

Global longitudinal strain (GLS) of the LV was measured by automate function imaging (AFI) at a frame rate of 65.2 ± 10.5 frames/sec. After selecting the optimal two-dimensional image, the timing of aortic valve closure was derived from the pulse wave Doppler of the aortic valve, and the three-point click method in three apical planes (apical four-chamber, two-chamber, and long axis view) was used. AFI non-invasively tracked and analyzed GLS based on the two-dimensional speckle tracking method and displayed the combined results of GLS of the three planes in a single bull’s eye summary. The mean value of GLS was calculated by dividing the sum of the GLS of each segment by 18 [10].

### 2.3. Statistical Analyses

The Statistical Package for Social Sciences, version 18.0 for Windows (SPSS Inc., Chicago, IL, USA) was used for the statistical analysis. Data are presented as percentages or mean ± standard deviation. The differences in the categorical variables were evaluated by using the chi-square test, and the continuous variables were compared by using the independent *t* test. Event-free survival rate was evaluated by using the Kaplan-Meier analysis, and event rates were compared by using the log-rank test. To identify the independent predictor of mortality, a multivariate Cox regression model was used for each of the cut-offs, with covariates that had *p* < 0.05 on univariate analysis. A *p* value of <0.05 was considered as statistically significant.

## 3. Results

### 3.1. Baseline Clinical Characteristics

During 5.4 years of clinical follow-up, death was developed in 137 patients (25.6%), and the mortality rate was not different between males (n = 73, 25.8%) and females (n = 64, 25.3%) (*p* = ns).

Comparisons of baseline characteristics by sex between the survived and the dead are summarized in Table 1.

In females, hypertension, diabetes, chronic kidney disease, and peripheral artery disease were significantly frequent in the dead than in the survived.

In males, the average age of the population was higher, and body weight and body mass index were significantly lower in the dead than in the survived group. Hypertension and chronic kidney disease were significantly more frequent in the dead than in the survived.

Hypertension and chronic kidney disease were significantly associated with mortality in both sexes.

### 3.2. Laboratory Findings

Comparisons of laboratory findings by sex between the survived and the dead are summarized in Table 2. The level of hemoglobin was significantly lower in death group in both sexes. Total cholesterol and HDL cholesterol were lower in female death group. Creatinine, hs-CRP, and NT-pro BNP were higher in the female death group. Triglyceride was higher in the death group than in the alive group of males. Other laboratory findings were not different between the groups.

### 3.3. Echocardiographic and Coronary Angiographic Findings

Comparisons of echocardiographic findings by sex between the survived and the dead are summarized in the Table 3.

LVEF was lower in death group in both sexes. LV end systolic dimension, RVSP, and E/E’ were higher in female death group. Male specific echocardiographic parameter was not shown in ICMP.

Coronary angiographic findings revealed involved vessel number was higher in the male death group (Table 4).

### 3.4. Medication

The differences of medication are summarized in Table 5. The user of beta blocker and angiotensin receptor blocker (ARB) was more frequent in female alive group. Nitrate was more used in female death group. Statin was frequently prescribed in male alive group compared with the death group or the female group.

### 3.5. Clinical Outcomes

A multivariate Cox regression model was used for each of the cut-offs, which defined from median value or normal limit with covariates that had *p* < 0.05 on univariate analysis. Low-ejection fraction was only an independent predictor of mortality in ICMP regardless of sex (HR 3.070 CI = 1.708–5.520 in female, HR 2.011, CI = 1.146–3.527 in male). Diabetes (HR 1.811, CI = 1.016–3.229), elevated e/e’ (HR 2.479, CI = 1.201–5.117), elevated pulmonary artery systolic pressure (HR 2.833, CI = 1.197–6.704), anemia (HR 1.860, CI = 1.025–3.373), beta blocker non-use (HR2.148, CI = 1.010–4.568), and angiotensin receptor blocker non-use (HR 2.100,CI = 1.137–3.881) were bad prognostic factors of long-term mortality in female, but use of beta blocker (HR 0.466, CI = 0.219–0.990) and angiotensin receptor blocker (HR 0.476, CI = 0.258–0.880) were good prognostic factors. Hypertension (HR 1.770, CI = 1.024–3.058), BMI (HR0.897, CI = 0.806–0.998), multi-vessel involved ICMP (HR 1.455, CI = 1.093–1.910), elevated creatinine (HR 2.188, CI = 1.225–3.908), and statin non-use (HR 3.475, CI = 1.989–6.071) were predictors of mortality in males, and statin (HR 0.600, CI = 0.260–1.387) was an improving factor of mortality in males with ICMP, independently (Figure 2).

Kaplan-Maier survival analysis showed a poorer outcome, which has multiple risk factors in ICMP (Figure 3). The progress was worse as the number of risk factors increased in both sexes. The fewer the predictors of mortality, the better the long-term prognosis.

## 4. Discussion 

In this present study, we investigated the sex-specific prognostic factors of long-term mortality in patients with elderly ICMP.

Firstly, regardless of sex, low ejection fraction was a significantly bad prognostic factor of mortality in ICMP.

Second, Diabetes, anemia, non-use of beta blocker, and non-use of ARB were female-specific risk factors in ICMP. Advanced diastolic dysfunction, such as elevated e/e’ and elevated RVSP, was also a female-specific predictor of mortality in elderly females, but not in males, in ICMP.

Third, hypertension, elevated Cr, and non-use of statin were predictors of mortality in males with ICMP independently. For improving long term survival in ICMP, it may be necessary to approach sex specifically, especially in elderly patients.

HF is a disease with a poor prognosis, and it appears frequently at the last phase of various heart diseases. Despite the advances in drug and device therapy for HF, the mortality rate in patients with HF remains high. It is comparable to that of the most common cancers, with <50% four-year survival [11]. One of the most prominent etiologies is IHD [12,13], ischemia exacerbation-progressed ICMP. ICMP is a morbid condition with a 10-year mortality rate of 60%. These patients have a multitude of comorbidities, including LV systolic dysfunction, impaired coronary hemodynamics, abnormal myocardial energetics, increased myocardial oxygen consumption, and altered myocardial lactate metabolism [14].

Elderly status is also related with many risk factors and comorbid conditions that may increase the risk of death. Therefore, elderly patients with ICMP were high risks in themselves, but there were some differences among them according to sex. Previous reports showed that sex differences in ICMP epidemiology depend on the age of the patient as the effect of sex on the outcome changes across the lifespan. In middle age, the rates of ICMP begin to increase in women, concomitant with the onset of menopause and loss of female sex hormone [15]. After middle age, event rates continually rise in women, with some reports of higher mortality in elderly women (85 years) compared with elderly men [16].

Traditional cardiovascular prognostic factors affect differently according to sex on long term mortality in this study. Decreased LV EF was only common prognostic factor on long term mortality in both sexes with ICMP. Diabetes was a female-specific, whereas hypertension was a male-specific predictor of long-term mortality in ICMP. Previous studies demonstrated that the excess risk of diabetes-related HF is significantly greater in women with diabetes than in men with diabetes [17]. Since women generally develop cardiovascular disease later in life than men, the age-adjusted relative risk is higher in women than in men, but some evidence suggests that diabetes confers a higher absolute risk in women than it does in men [18]. Prognosis is much worse among those women with diabetes than among women without diabetes, although the prognoses for women and men with diabetes are similar [19]. Hypertension is more prevalent among females than males with HF, as was shown in previous reports. This could be explained by a higher augmentation index between peripheral and central blood pressure in women compared with men, which contributes to greater end organ damage, including LV hypertrophy [20]. In our elderly population, incidence was similar with previous reports, and mortality prediction was male-specific.

In this study, anemia was the female-specific risk factor of mortality in ICMP. Anemia can cause worse symptoms and prognosis in heart failure. It may worsen ischemia in patients who already have ICMP, and uncontrolled ischemia can deteriorate prognosis in this group of patients.

Even though systolic function was a common risk factor of mortality, diastolic dysfunction was a susceptibility for long-term mortality in female rather than male patients with ICMP. Generally, diastolic dysfunction, especially relaxation abnormality, was regarded as phenotype of cardiac aging, since, in the case of our elderly population, diastolic dysfunction was considered somewhat natural. According to ischemic cascade flow, diastolic dysfunction was located in front of systolic dysfunction in ischemic cascade flow [21]. From this data, females may be friable for diastolic function in ICMP.

Angiotensin converting enzyme inhibitors/ARB, beta blocker, diuretics, antiplatelets, and statins are included in the medical therapies used in patients with ICMP. Selected cases consider vasodilators, ivabradine, aldosterone antagonists, and/or the angiotensin receptor neprilysin inhibitors (ARNI) nowadays. Because our population consists of chronic patients who were diagnosed several years ago, new drug data were not included in this study, unfortunately.

There are currently no therapeutic guidelines regarding sex-based treatments. A previous study of hospitalized chronic HF patients revealed that women were equally likely to receive diuretics, but less likely to receive vasodilators or to be treated with evidence-based therapy than men [22].

Beta blockers may improve the function of viable, but hibernating, myocardium by reducing myocardial oxygen consumption and increasing diastolic perfusion [23].

Inhibition of renin angiotensin system decreases specific neurohumoral activation in chronic HF patients, at least partly, by slowing disease progression. From this study, beta blockers and ARB were more important for long-term prognosis in females, which suggests that autonomic nerve system and neurohormonal regulation were more sensitive in females than in males.

Statin use is a critical factor of long term survival in ICMP, especially in males, but not in females. All individuals with CAD have high risk of cardiovascular events and should be treated with statins according to the recommendations of the ESC/European Atherosclerosis Society guidelines for the management of dyslipidemia, regardless of low-density lipoprotein cholestestol (LDL-C) levels [24]. The mechanism was unclear, but statins were essential for male ICMP for improving long-term outcome.

### Study Limitations

There are several potential limitations in this study. Firstly, the present study was a single-center, retrospective, observational study with a relatively small number of patients and thus has all limitations of retrospective analysis, including selection bias in study subjects. Secondly, the duration of ICMP was not exactly clear in the present study, as it may influence long-term MACE. Thirdly, the present study is mostly based on previous diagnosed ICMP. Therefore, we did not consider new medications, such as ARNI or SGLT2I. Recently diagnosed and recovered ICMP treated with ARNI takes only a relatively small portion of this study. Thus, we have no long-term data about ICMP related to ARNI.

## 5. Conclusions

Despite these potential limitations, the results of the present study demonstrated that low-ejection fraction was the only independent predictor of mortality in ICMP, regardless sex.

Diabetes and impaired diastolic function, non-use of beta blockers, and ARB were female-specific predictors of long-term mortality in patients with ICMP.

Hypertension and non-use of statins were male-specific predictors of long-term mortality in patients with ICMP.

For improving long term survival in ICMP, it may be necessary to approach sex specifically when we meet with elderly HFs with ICMP. Prospective long-term study for effect and mechanism of medication, including new drugs, will give us more customized information in the near future.

## Figures and Tables

**Figure 1 jcm-12-02012-f001:**
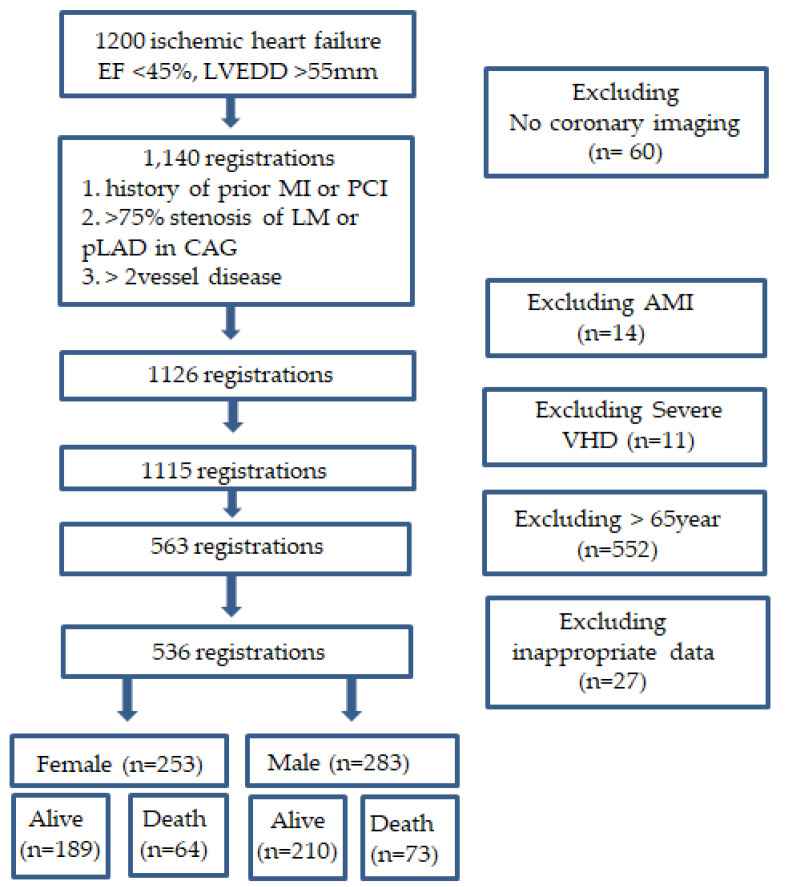
Patients’ inclusion flow chart. EF: ejection fraction; LVEDD: left ventricular ejection fraction; MI: myocardial infarction; PCI: percutaneous coronary intervention; LM: left main artery; pLAD: proximal left anterior descending; CAG: coronary angiography; AMI: acute myocardial infarction; VHD: valvular heart disease.

**Figure 2 jcm-12-02012-f002:**
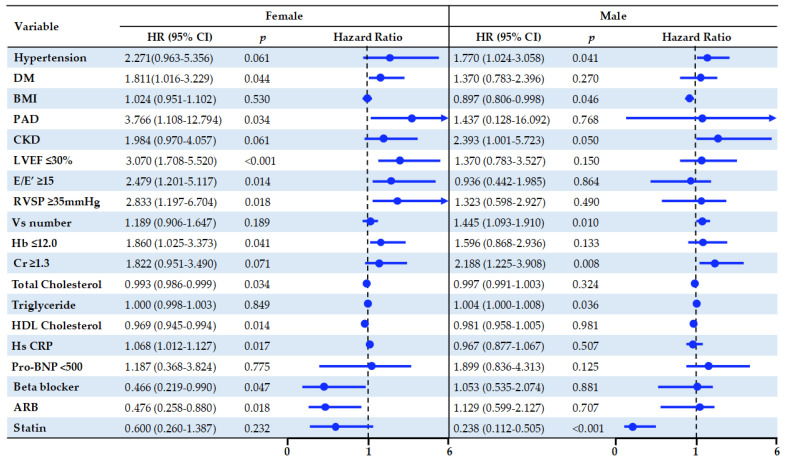
Forest plots of HRs for independent predictors identified using multivariate Cox proportional hazards regression analysis in elderly ICMP. HR: hazard ratio; CI: confidence interval; DM: diabetes mellitus; BMI: body mass index; PAD; peripheral arterial disease; CKD: chronic kidney disease; LVEF: left ventricular ejection fraction; RVSP: right ventricular systolic pressure; Vs: vessel; Hb: hemoglobin; Cr: creatinine; HDL: high density lipoprotein; HS CRP: high sensitivity C reactive protein; Pro BNP: N-terminal pro B type natriuretic peptide; ARB: angiotensin receptor blocker.

**Figure 3 jcm-12-02012-f003:**
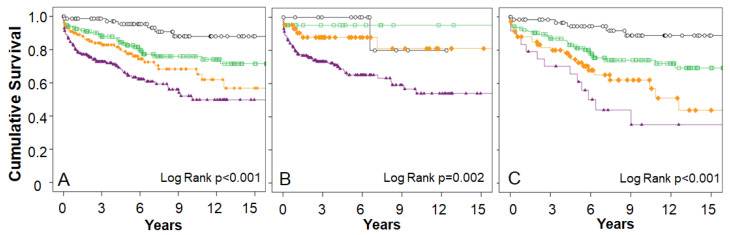
Kaplan-Maier curves according to number of predictors of mortality in elderly ICMP. (**A**): overall, (**B**): female, (**C**): Male, Risk factor number: ○: 0, □: 1, ♦: 2, ▲: ≥3.

**Table 1 jcm-12-02012-t001:** Baseline characteristics.

	Female (N = 253)			Male (N = 283)		
	Alive (n = 189)	Death (n = 64)	*p*	Alive (n = 210)	Death (n = 73)	*p*
Age	78.6 ± 6.8	80.5 ± 7.9	0.057	75.9 ± 6.6	79.2 ± 7.2	<0.001
Height (cm)	152.9 ± 5.8	153.3 ± 8.3	0.688	167.1 ± 6.3	166.0 ± 5.7	0.203
Weight (kg)	53.9 ± 9.1	55.4 ± 11.4	0.294	67.7 ± 11.0	64.5 ± 7.9	0.031
BMI, kg/m^2^	23.0 ± 3.4	23.3 ± 4.7	0.531	24.3 ± 3.2	23.4 ± 2.3	0.045
HTN (%)	147 (77.8)	57 (89.1)	0.038	103 (49.0)	46 (63.0)	0.027
DM (%)	87 (46.0)	39 (60.9)	0.030	63 (30.0)	27 (37.0)	0.169
Dyslipidemia (%)	25 (13.2)	9 (14.1)	0.296	9 (4.3)	2 (2.7)	0.428
Smoking (%)	20 (10.6)	6 (9.4)	0.503	150 (71.4)	51 (69.9)	0.455
Family Hx (%)	15 (7.9)	4 (6.3)	0.449	9 (4.3)	3 (4.1)	0.625
PAD (%)	5 (2.6)	6 (9.4)	0.034	2 (1.0)	1 (1.4)	0.594
CKD (%)	25 (13.2)	15 (23.4)	0.048	13 (8.2)	10 (13.7)	0.044
CVA (%)	22 (11.6)	10 (15.6)	0.275	23 (11.0)	10 (13.7)	0.335
SBP (mmHg)	126.1 ± 23.5	123.9 ± 24.3	0.515	125.9 ± 19.7	125.2 ± 22.0	0.785
DBP (mmHg)	78.6 ± 15.4	77.4 ± 13.7	0.569	78.5 ± 13.2	79.0 ± 14.2	0.816
HR (/min)	80.7 ± 15.5	82.6 ± 20.5	0.436	80.0 ± 18.4	82.3 ± 18.4	0.351
AF (%)	26 (13.8)	5 (7.8)	0.153	32 (15.2)	19 (26.0)	0.052

BMI: body mass index; HTN: hypertension; DM: diabetes mellitus; Hx: history; PAD: peripheral arterial disease; CKD: chronic kidney disease; CVA: cerebrovascular accident; SBP: systolic blood pressure; DBP: diastolic blood pressure; HR: heart rate; AF: atrial fibrillation.

**Table 2 jcm-12-02012-t002:** Laboratory findings.

	Female (N = 253)			Male (N = 283)		
	Alive (n = 189)	Death (n = 64)	*p*	Alive (n = 210)	Death (n = 73)	*p*
WBC (mg/dL)	9120 ± 3.6	9860 ± 4.5	0.184	100.3 ± 4.7	104.10 ± 4.3	0.583
Hb (g/dL)	11.8 ± 1.7	11.0 ± 1.6	<0.001	13.5 ± 1.9	13.0 ± 2.2	0.040
Platelet	243.10 ± 68.3	242.45 ± 72.2	0.949	214.35 ± 61.6	215.81 ± 75.1	0.869
Albumin (mg/dL)	3.63 ± 0.5	3.56 ± 0.5	0.274	3.85 ± 0.5	3.71 ± 0.5	0.051
Glucose (g/dL)	182.5 ± 117.1	199.2 ± 98.0	0.307	158.4 ± 78.1	156.6 ± 79.8	0.872
Cr (mg/dL)	1.15 ± 1.1	1.57 ±1.9	0.030	1.27 ± 1.3	1.44 ± 1.4	0.299
TC (mg/dL)	180.0 ± 49.1	165.0 ± 42.8	0.033	170.8 ± 43.3	164.7 ± 48.1	0.325
LDLC (mg/dL)	113.7 ± 39.6	103.8 ± 40.4	0.093	107.0 ± 38.8	100.3 ± 38.0	0.154
TG (mg/dL)	120.5 ± 114.9	123.6 ± 94.9	0.849	110.7 ± 59.5	134.0 ± 102.6	0.025
HDLC (mg/dL)	43.3 ± 12.5	38.9 ± 10.3	0.013	43.6 ± 11.9	40.9 ± 13.0	0.117
CK-MB (ng/mL)	73.7 ± 132.3	63.4 ± 116.1	0.582	114.2 ± 181.0	108.6 ± 152.9	0.824
Tn I (ng/mL)	35.3 ± 52.5	30.1 ± 51.1	0.493	39.8 ± 71.5	47.1 ± 76.9	0.464
HbA1c	6.71 ± 1.3	7.05 ± 1.4	0.111	6.76 ± 1.3	6.68 ± 1.4	0.753
Hs CRP (mg/dL)	2.37 ± 4.7	4.40 ± 6.3	0.011	1.93 ± 3.4	1.62 ± 2.3	0.508
NT-pro BNP (pg/mL)	7724.2 ± 9466	15,237 ± 13,008	<0.001	4553 ± 8006	5068 ± 8114	0.714

WBC: white cell count; Hb: hemoglobin; Cr: creatinine; TC: total cholesterol; LDLC: low density lipoprotein cholesterol; TG: triglyceride; HDLC: high density lipoprotein cholesterol; CK-MB: creatin kinase-MB; Tn-I: troponine-I; Hs CRP: high sensitive C reactive protein; NT-pro BNP: N terminal-pro B type natriuretic peptide.

**Table 3 jcm-12-02012-t003:** Echocardiographic findings.

	Female (N = 253)			Male (N = 283)		
	Alive (n = 189)	Death (n = 64)	*p*	Alive (n = 210)	Death (n = 73)	*p*
LVEDD (mm)	58.6 ± 6.3	59.0 ± 6.2	0.633	61.7 ± 6.1	61.2 ± 6.5	0.591
LVESD (mm)	46.7 ± 8.1	49.7 ± 7.6	0.012	50.0 ± 7.7	50.1 ± 8.7	0.978
EF (%)	33.6 ± 9.1	28.9 ± 8.4	0.009	35.4 ± 8.8	31.3 ± 10.4	0.001
GLS (%)	−8.5 ± 2.8	−8.3 ± 3.2	0.862	−7.3 ± 2.4	−9.6 ± 3.2	0.201
LAD (mm)	43.9 ± 7.0	44.0 ± 7.4	0.907	44.7 ± 7.2	44.5 ± 6.9	0.785
E wave (m/s)	0.79 ± 0.33	0.98 ± 0.35	<0.001	0.78 ± 0.29	0.77 ± 0.27	0.765
A wave (m/s)	0.86 ± 0.28	0.79 ± 0.25	0.087	0.74 ± 0.29	0.79 ± 0.26	0.274
DT (ms)	194.5 ± 84.3	146.7 ± 44.8	<0.001	185.9 ± 89.0	185.0 ± 82.6	0.948
E’ wave (m/s)	0.04 ± 0.01	0.04 ± 0.01	0.779	0.05 ± 0.17	0.04 ± 0.18	0.765
RVSP (mmHg)	43.1 ± 13.8	51.3 ± 15.3	<0.001	43.4 ± 13.6	45.0 ± 13.6	0.490
E/E’ ratio	20.8 ± 11.2	25.9 ± 12.1	0.004	20.8 ± 11.2	19.6 ± 7.8	0.539

LVEDD: left ventricular end diastolic dimension; LVESD: left ventricular end systolic dimension; EF: ejection fraction; GLS: global longitudinal strain; LAD: left atrial dimension; DT: deceleration time; RVSP: right ventricular systolic pressure.

**Table 4 jcm-12-02012-t004:** Coronary angiographic findings.

	Female (N = 253)			Male (N = 283)		
	Alive (n = 189)	Death (n = 64)	*p*	Alive (n = 210)	Death (n = 73)	*p*
Admission No.	2.97 ± 2.5	3.02 ± 2.5	0.896	3.49 ± 2.9	3.95 ± 3.6	0.285
PCI No.	1.60 ± 0.9	1.64 ± 1.4	0.819	1.51 ± 1.1	1.37 ± 0.97	0.333
Vessel No.	2.04 ± 0.9	2.22 ± 1.0	0.189	1.86 ± 0.9	2.21 ± 1.0	0.009
LM (%)	20 (10.6)	9 (14.1)	0.278	21 (10.0)	13 (17.8)	0.057
LAD (%)	153 (81.0)	55 (85.9)	0.292	153 (72.9)	60 (82.2)	0.055
RCA (%)	108 (57.1)	42 (65.6)	0.145	116 (55.2)	46 (63.0)	0.135
LCX (%)	101 (53.4)	39 (60.9)	0.172	103 (49.0)	40 (54.8)	0.228
Multi vessel (%)	124 (65.6)	47 (73.4)	0.132	127 (60.5)	50 (68.5)	0.341
CABG (%)	13 (6.9)	9 (14.1)	0.073	27 (12.9)	8 (11.0)	0.433

PCI: percutaneous coronary intervention; LM: left main; LAD: left anterior descending; RCA: right coronary artery; LCX: left circumflex artery; CABG: coronary artery bypass graft.

**Table 5 jcm-12-02012-t005:** Medication for groups.

	Female (N = 253)			Male (N = 283)		
	Alive (n = 189)	Death (n = 64)	*p*	Alive (n = 210)	Death (n = 73)	*p*
Aspirin (%)	165 (87.3)	58 (90.1)	0.598	197 (93.8)	62 (84.9)	0.304
Plavix (%)	142 (75.1)	51 (79.7)	0.242	157 (74.8)	51 (69.9)	0.547
ACEI (%)	33 (17.5)	11 (17.2)	0.494	18 (8.6)	6 (8.2)	0.540
ARB (%)	117 (61.9)	32 (54.7)	0.014	142 (67.6)	47 (64.3)	0.419
Loop diuretics (%)	101 (53.4)	38 (59.4)	0.413	116 (55.2)	36 (49.3)	0.474
Beta blocker (%)	145 (76.7)	45 (70.3)	0.039	156 (74.3)	50 (68.5)	0.515
CCB (%)	22 (11.6)	6 (9.4)	0.348	47 (22.4)	12 (16.4)	0.283
Statin (%)	147 (77.8)	49 (76.6)	0.164	186 (88.6)	47 (64.4)	<0.001
Herben (%)	15 (7.9)	4 (6.3)	0.410	33 (15.7)	9 (12.3)	0.415
Nitrate (%)	44 (23.3)	24 (37.5)	0.034	92 (43.8)	31 (42.5)	0.384
Vastinan (%)	23 (12.2)	6 (9.4)	0.311	15 (7.1)	6 (8.2)	0.371
Aldactone (%)	100 (52.9)	37 (57.8)	0.431	110 (52.4)	36 (49.3)	0.458

ACEI: angiotensin converting enzyme inhibitor; ARB: angiotensin receptor blocker; CCB: calcium channel blocker.

## Data Availability

No new data were created or analyzed in this study. Data sharing is not applicable to this article.

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
