# Peer review of "Sex-Specific Predictors of Long-Term Mortality in Elderly Patients with Ischemic Cardiomyopathy"

_jcm, 2023, doi:10.3390/jcm12052012_

Round 1
Reviewer 1 Report
1. Please include the interquartile range when mentioning HR in the abstract and the rest of the manuscript.
2. Consider changing line 31, “heart attack or coronary artery disease”, to “acute ischemic syndrome”.
3. Why did you decide that the inclusion criteria for LVEF were <45 % instead of 40 %, as we usually use for heart failure to reduce ejection fraction?
4. Consider including a flowchart for inclusion and exclusion criteria, which will be helpful.
5. In line 99, you mentioned the independent predictor of chemotherapy-induced LVD. Could you explain?
6. You used logistic regression. Why not use Cox regression?
7. Could you include or discuss that you performed confounding analysis and diagnostics of the regression model?
8. In figure 1, could you include abbreviations or mention where the reader can find them (another table)?
9. In Figure 1, change the follow-up to years instead of days.
10. In part referring to limitations, it mentions “Firstly” twice, in lines 237 and 240.
Author Response
I really appreciate your thoughtful comments. I made some changes to accommodate your opinions.
- Please include the interquartile range when mentioning HR in the abstract and the rest of the manuscript.
=> I inserted confidential interval in abstract and Results part.
- Consider changing line 31, “heart attack or coronary artery disease”, to “acute ischemic syndrome”.
=> I changed the phrase according to your comment.
- Why did you decide that the inclusion criteria for LVEF were <45 % instead of 40 %, as we usually use for heart failure to reduce ejection fraction?
=> As your point, current definition of ICMP is LVEF<40%, generally. In this study, the patients were diagnosed ICMP from 2007 to 2017. A guideline from British Society of Echocardiography published in 2017, the dilated cardiomyopathy was defined as LV dilatation (>112% corrected to BSA and age) with reduced function (FS<25% and/or LVEF <45%). Ischemic DCMP criteria was not different at that time. Therefore, for the purpose of detecting predictor of mortality, we included patients with ejection fraction ranging from 40-45% (n=53).
Reference : Mathew T, Williams L, Navaratnam G,et al. Diagnosis and assessment of dilated cardiomyopathy: a guideline protocol from the British Society of Echocardiography. Echo Res Pract 2017 Jun;4(2):G1-G13.
- Consider including a flowchart for inclusion and exclusion criteria, which will be helpful. => We made figure 1 as a patient inclusion flow chart.
- In line 99, you mentioned the independent predictor of chemotherapy-induced LVD. Could you explain?
=> The phrase was a typo. Sorry for my mistake. I corrected it to mortality.
- You used logistic regression. Why not use Cox regression?
=> In methodology part, it was my mistake to write statistical analyses according to previous my reports. To identify the independent predictor of mortality, a multivariate Cox regression model was used for each of the cut-offs, with covariates that had P < 0.05 on uni-variate analysis. We revised some sentences in method section.
- Could you include or discuss that you performed confounding analysis and diagnostics of the regression model?
=> We inserted in result session.
- In figure 1, could you include abbreviations or mention where the reader can find them (another table)? => I inserted abbreviations for each table.
- In Figure 1, change the follow-up to years instead of days.
=> We changed the follow-up to years.
- In part referring to limitations, it mentions “Firstly” twice, in lines 237 and 240.
=> Thank you for pointing it out. I erased the repeated sentence.

Reviewer 2 Report
The paper is well written and the results are very interesting and original mainly concerning the sex difference aspect of the work. there is no doubt that a larger number of patients would make the work more solid. It is suggested in some parts to use the word sex-dependent instead of gender since this latter is less specific and includes social and cultural aspects which are not taken into consideration in the present study. The authors should indicate clearly the inclusions and exclusion criteria such as race, smoking, etc.,
Author Response
I really appreciate your thoughtful comments. I made some changes such as adding flow chart of inclusion criteria to accommodate your opinions.
Thank you.

Round 2
Reviewer 1 Report
None